# Elevated TAT in COVID-19 Patients with Normal D-Dimer as a Predictor of Severe Respiratory Failure: A Retrospective Analysis of 797 Patients

**DOI:** 10.3390/jcm11010134

**Published:** 2021-12-27

**Authors:** Yuichiro Takeshita, Jiro Terada, Yasutaka Hirasawa, Taku Kinoshita, Hiroshi Tajima, Ken Koshikawa, Toru Kinouchi, Yuri Isaka, Yu Shionoya, Atsushi Fujikawa, Yuji Tada, Chiaki Nakaseko, Kenji Tsushima

**Affiliations:** 1Department of Pulmonary Medicine, International University of Health and Welfare Narita Hospital, 852 Hatakeda, Narita 286-8520, Japan; y-takeshita@iuhw.ac.jp (Y.T.); yahirasaw@iuhw.ac.jp (Y.H.); tkino@iuhw.ac.jp (T.K.); htajima@iuhw.ac.jp (H.T.); koshi6488@yahoo.co.jp (K.K.); tkinouchi@iuhw.ac.jp (T.K.); lily.well.slope@iuhw.ac.jp (Y.I.); yshymta616@gmail.com (Y.S.); atsuf51202@gmail.com (A.F.); ytada25@iuhw.ac.jp (Y.T.); ktsushima@iuhw.ac.jp (K.T.); 2Department of Respirology, Graduate School of Medicine, Chiba University, 1-8-1 Inohana, Chiba 260-8670, Japan; 3Department of Hematology, International University of Health and Welfare Narita Hospital, 852 Hatakeda, Narita 286-8520, Japan; cnakaseko@iuhw.ac.jp

**Keywords:** COVID-19, D-dimer, TAT, coagulation, fibrinolysis, disseminated intravascular coagulation, respiratory failure

## Abstract

Although previous studies have revealed that elevated D-dimer in the early stage of coronavirus 2019 (COVID-19) indicates pulmonary intravascular coagulation, the state of coagulation/fibrinolysis disorder with normal D-dimer is unknown. The study aimed to investigate how coagulation/fibrinolysis markers affect severe respiratory failure in the early stage of COVID-19. Among 1043 patients with COVID-19, 797 patients were included in our single-center retrospective study. These 797 patients were divided into two groups, the normal D-dimer and elevated D-dimer groups and analyzed for each group. A logistic regression model was fitted for age, sex, body mass index (BMI) ≥ 30 kg/m^2^, fibrinogen ≥ 617 mg/dL, thrombin-antithrombin complex (TAT) ≥ 4.0 ng/mL, and plasmin-alpha2-plasmin inhibitor-complex (PIC) > 0.8 µg/mL. A multivariate analysis of the normal D-dimer group demonstrated that being male and TAT ≥ 4.0 ng/mL significantly affected severe respiratory failure. In a multivariate analysis of the elevated D-dimer group, BMI ≥ 30 kg/m^2^ and fibrinogen ≥ 617 mg/dL significantly affected severe respiratory failure. The elevated PIC did not affect severe respiratory failure in any group. Our study demonstrated that hypercoagulation due to SARS-CoV-2 infection may occur even during a normal D-dimer level, causing severe respiratory failure in COVID-19.

## 1. Introduction

Globally, as of 19 November 2021, a total of 255,324,963 confirmed cases of coronavirus 2019 (COVID-19), including 5,127,696 deaths, has been reported to the World Health Organization (WHO) [1]. Recent research has revealed that COVID-19 is frequently associated with coagulation dysfunction, especially in severe cases, and is a risk for death [2].

Although COVID-19, which can cause a critically severe condition, is associated with coagulation disorders, some aspects regarding the pathophysiology of coagulation dysfunction, which cannot be explained solely by classical acute respiratory distress syndrome (ARDS) or disseminated intravascular coagulation (DIC), are still unknown. In a postmortem report of COVID-19, diffuse alveolar damage, thrombin formation, and bleeding in the microvasculature of the lungs were observed, suggesting that the mechanism of coagulation dysfunction was different from that of classical ARDS [3]. While most patients with COVID-19 do not meet the criteria for the usual forms of DIC, DIC occasionally complicates COVID-19 (0.6% in surviving cases; 71.4% in lethal cases), and it is difficult to survive once DIC develops [2,4]. A previous report has suggested that strong local fibrin formation in the lungs causes pulmonary intravascular coagulation, particularly in the early stages of COVID-19 [4,5,6].

Elevated D-dimer levels have been revealed as an essential indicator of coagulation dysfunction in COVID-19 [2,5,7]. A previous report has suggested that severe pulmonary intravascular coagulation increases D-dimer levels, whereas a decrease in fibrinogen in classical DIC is not observed [6]. Furthermore, another report has mentioned that elevated D-dimer levels in the early stages of COVID-19 are a characteristic indicator of pulmonary intravascular coagulation, partly because D-dimer concentrations above 1 µg/mL were associated with an 18-fold increase in the odds ratio for fatal outcomes [5,6].

Thus, is there any way to estimate the state or degree of abnormal coagulation disorder, if the value of D-dimer is normal? Generally, DIC associated with severe infections and sepsis does not show fibrinolytic activity despite a high level of thrombus formation and often results in fibrinolytic-suppressing DIC. Therefore, thrombin-antithrombin complex (TAT), as a coagulation activation marker, increases, but plasmin-alpha2-plasmin inhibitor-complex (PIC), which is a fibrinolytic activation marker, increases only mildly, and the increase in D-dimer may not be remarkable [8]. Thus, even when the D-dimer value is not extremely high, it may be possible to evaluate the balance between coagulation and fibrinolytic activation and predict DIC by observing TAT and PIC values. 

In this study, we retrospectively studied how coagulation markers at admission affect severe respiratory failure after admission among patients admitted with COVID-19.

## 2. Materials and Methods

### 2.1. Ethical Approval

All study procedures were conducted according to the standards of the Ethical Review Board of the International University of Health and Welfare (approval number 20-Nr-101; 22 February 2021 approved) and conformed to the 1964 Declaration of Helsinki and its subsequent amendments or comparable ethical standards. The requirement for informed consent was waived by the Ethics Committee because this retrospective analysis was limited to preexisting data collected as part of the standard of care by respiratory physicians. Furthermore, data anonymization and privacy were protected.

### 2.2. Study Design and Subjects

This single-center retrospective study was investigated 1043 adult patients with COVID-19, who were admitted to the International University of Health and Welfare Narita Hospital between March 2020 and September 2021. Infection of COVID-19 was confirmed using quantitative reverse-transcription polymerase chain reaction assay. Among 1043 patients with COVID-19 admitted to our hospital during this period, 241 patients were excluded due to insufficient data on coagulation and fibrinolytic system factors, and five patients were excluded because of duplicate data due to retransfer. Therefore, 797 patients were included in our study, who were subjected to a three-step analysis. First, the patients were divided into two groups, a severe group with severe respiratory failure and a non-severe group and analyzed for their characteristics (analysis 1). Second, the patients were divided into two another groups: a group with D-dimer < 1 μg/mL (i.e., normal D-dimer group, *n* = 589) and a group with D-dimer ≥ 1 μg/mL (i.e., elevated D-dimer group, *n* = 208). Seventy-four patients (12.6%) of the normal D-dimer group and 52 patients (25.0%) of the elevated D-dimer group were classified in the severe group. Then, the factors associated with severe respiratory failure were analyzed for the normal D-dimer (analysis 2) and elevated D-dimer groups (analysis 3). The study flowchart is shown in Figure 1.

### 2.3. Clinical Assessment

The hospital’s electronic medical records were used to extract data during hospitalization, such as symptoms, vital signs, electrocardiography findings, peripheral capillary oxygen saturation (SpO_2_), oxygen demand, laboratory test results, computed tomography (CT) scans, and patient characteristics, including age (in years), sex, and body mass index (BMI; in kg/m^2^). All laboratory test results were measured on admission. Data on coagulation and fibrinolytic system factors, including fibrinogen, D-dimer, TAT, and PIC, were routinely collected on the admission of COVID-19 patients in our hospital. CT findings were evaluated for pneumonia by two skilled operators (one radiologist and one pulmonologist) who were blinded to the clinical history. The data on SpO_2_ ≤ 93% on room air at sea level, conventional oxygen therapy (COT), high-flow nasal cannula (HFNC) treatment, and mechanical ventilation (MV) administration were evaluated.

### 2.4. Definition of Severe Respiratory Failure, Severe Group, and Non-Severe Group

We defined patients who required the use of a high-flow nasal cannula (HFNC) or MV after hospitalization as severe respiratory failure patients. We classified patients with severe respiratory failure and all other patients in the severe and non-severe groups, respectively.

### 2.5. The Setting of the Cutoff Value of Coagulation Factors

Prothrombin time (PT) > 13 s, activated partial thromboplastin time (APTT) > 40 s, D-dimer ≥ 1.0 μg/mL, TAT ≥ 4.0 ng/mL, and PIC > 0.8 μg/mL were set as abnormal values. Also, fibrinogen ≥ 617 mg/dL was set as an abnormal value according to the previous report by Di Mico et al. [9].

### 2.6. Statistical Analysis

Summary statistics were calculated for baseline variables using the mean (± standard deviation (SD)), frequency distributions, or proportions. The differences between severe and non-severe groups were analyzed. For continuous variables, we first compared the mean values (±SD) and quartiles between the two groups. Subsequently, the Kolmogorov–Smirnov test (2-sided) and Shapiro–Wilk test were used to test normality, and homoscedasticity was further tested using the F-test. The Welch *t*-test and Mann–Whitney U test were performed according to the data distribution. For continuous variables, such as age and BMI, we first compared the mean values (±SD) and quartiles between the two groups. The Fisher exact test was used to determine the significance of differences in the proportions of the groups. The key characteristics of the variables were studied. In an analysis with 589 patients presented with D-dimer < 1 µg/mL and an analysis with 208 patients presented with D-dimer < 1 µg/mL, a logistic regression model was fitted with age, sex, BMI ≥ 30 kg/m^2^, fibrinogen ≥ 617 mg/dL, TAT ≥ 4 ng/mL, and PIC > 0.8 µg/mL. A *p* value < 0.05 was considered statistically significant. All statistical analyses were performed using EZR (Saitama Medical Center, Jichi Medical University, Saitama, Japan), which is a graphical user interface for R (a modified version of R commander designed to add statistical functions frequently used in biostatistics) [10].

### 2.7. Patient and Public Involvement

No patients were involved in determining the research questions, outcome measures, or study design. There was no patient input in the interpretation or writing of the results.

## 3. Results

### 3.1. Patient Characteristics (Analysis 1)

Table 1 presents clinical characteristics of 797 patients in the study cohort (non-severe group, *n* = 671, and severe group, *n* = 126). Based on univariate analysis, the values of BMI, APTT, fibrinogen, and TAT and the proportion of males were significantly higher in the severe group than in the non-severe group. No significant difference was observed in age, PT, D-dimer, and PIC value between the two groups.

### 3.2. Percentage of Patients in the Severe Group with Abnormal Values of Coagulation/Fibrinolytic Markers

First, we calculated the proportion of patients with abnormal values of coagulation/fibrinolysis markers out of all 126 severe patients (Figure 2a). Among these markers, elevated TAT showed the highest proportion. Then, the severe group was divided into two groups, the D-dimer normal group, and the D-dimer elevated group, and the same calculation was performed (Figure 2b,c). In both the normal D-dimer group and the elevated D-dimer group, patients with elevated TAT showed the highest proportion in the severe group. These results suggest that TAT affects severe respiratory failure caused by COVID-19.

### 3.3. Respiratory Status on Admission When D-Dimer Is Divided into Two Groups

To objectively evaluate respiratory status on admission, we first analyzed the proportion of patients who showed pneumonia on admission dividing them into two groups, the normal D-dimer group, and the elevated D-dimer group. Then, each variable presenting respiratory status, i.e., SpO_2_ ≤ 93% on room air at sea level, COT, HFNC, and MV was similarly analyzed (Table 2). The proportion of pneumonia was significantly higher in the elevated D-dimer group at any severity. This result suggests that the elevated D-dimer on admission itself reflects respiratory failure, that is, the advanced stage of COVID-19.

### 3.4. Analysis of Patients with Normal D-Dimer (Analysis 2)

Table 3 presents the clinical characteristics of the 589 patients in the normal D-dimer group (non-severe group, *n* = 515, and severe group, *n* = 74) by univariate analysis. BMI, fibrinogen, D-dimer, and TAT values and the proportion of males were significantly higher in the severe group than in the non-severe group. No significant difference was observed in age and PIC between the two groups.

Table 4 shows the analysis of factors that affect severe respiratory failure in the normal D-dimer group. Factors such as age, male sex, BMI ≥ 30 kg/m^2^, fibrinogen ≥ 617 mg/dL, TAT ≥ 4.0 ng/mL, and PIC > 0.8 μg/mL were analyzed. In the univariate analysis, male sex, fibrinogen ≥ 617 mg/dL, and TAT ≥ 4.0 ng/mL significantly affected severe respiratory failure. Furthermore, in the multivariate analysis, male sex and TAT ≥ 4.0 ng/mL significantly affected severe respiratory failure.

### 3.5. Analysis of Patients with Elevated D-Dimer (Analysis 3)

Table 5 presents clinical characteristics of 208 patients in the elevated D-dimer group (non-severe group, *n* = 156, and severe group, *n* = 52) by the univariate analysis. BMI and fibrinogen were significantly higher in the severe group than in the non-severe group. No significant difference was observed in age, D-dimer, TAT, and PIC between the two groups.

Table 6 shows the analysis of factors that affect severe respiratory failure in the elevated D-dimer group. Factors such as age, male sex, BMI ≥ 30 kg/m^2^, fibrinogen ≥ 617 mg/dL, TAT ≥ 4.0 ng/mL, and PIC > 0.8 μg/mL were analyzed. In the univariate analysis, BMI ≥ 30 kg/m^2^ and fibrinogen ≥ 617 mg/dL significantly affected severe respiratory failure. Furthermore, in the multivariate analysis, BMI ≥ 30 kg/m^2^ and fibrinogen ≥ 617 mg/dL significantly affected severe respiratory failure.

## 4. Discussion

The present study revealed several findings. First, an elevated TAT within normal D-dimer, which reflects coagulation activity, was an influential coagulation factor in severe respiratory failure caused by COVID-19 compared with fibrinogen and PIC. Second, compared with TAT and PIC, higher fibrinogen with elevated D-dimer, which reflects a disruption of the coagulation/fibrinolysis system due to remarkable inflammation, was also an influential coagulation factor in severe respiratory failure. Third, an elevated PIC, which reflects fibrinolytic activity, did not affect severe respiratory failure even in the elevated D-dimer group, which should be predominant in the fibrinolytic system.

The results of analysis 2 (Table 4) demonstrated that an elevated TAT with a normal D-dimer might be associated with severe respiratory failure in COVID-19. The result suggests that significant thrombin activity occurs even in the early stage of the disease, indicating that it can cause severe respiratory failure. The cause of excessive thrombin activity in COVID-19 may be excessive tissue factor (TF), a major source of coagulation activation, from vascular endothelial cells due to viral infection. It is known that the enhanced cytokine production during viral infection stimulates procoagulant reactions with increased TF expression [11,12]. From these facts, the hypothesis emerges that a large amount of TF is released from vascular endothelial cells by cytokine release due to severe acute respiratory syndrome coronavirus-2 (SARS-CoV-2) infection, resulting in increased thrombin activity.

As shown in the results of analysis 3 (Table 6), the higher value of fibrinogen with elevated D-dimer reflects the disruption of the coagulation/fibrinolysis system due to remarkable inflammation. Di Micco et al. reported that a cutoff value of 617 mg/dL has a sensitivity and specificity of 76% and 79%, respectively, in identifying COVID-19 patients with ARDS [9]. Ranucci M et al. pointed out that there is an association between fibrinogen and interleukin-6 (IL-6) in COVID-19 cases complicated with ARDS [13]. These previous reports suggest that numerous fibrinogens are produced by significant intravascular inflammation caused by viral infection. Likewise, several studies have mentioned the prognosis of COVID-19 by D-dimer levels on admission. Although there are multiple reports that D-dimer affects the severity of COVID-19, the cutoff value is not significantly high [14,15,16]. This suggests that the state of suppressed-fibrinolytic-type DIC continues even when D-dimer is elevated.

Then, why does a significant coagulation activity in COVID-19 affect respiratory failure? Anti-inflammatory and antithrombotic disruption of vascular endothelial cells may occur locally in the lungs. According to autopsy cases, a high amount of SARS-CoV-2 RNA is concentrated in the lungs, and the angiotensin-converting enzyme 2 (ACE2) receptor, which plays an important role in host cell invasion by SARS-CoV-2, is highly expressed in the lungs [17]. This suggests that ACE2-receptor-mediated viral infection expressed in vascular endothelial cells of pulmonary microvessels cause an increased thrombotic activity locally in the lungs. The original functions of vascular endothelial cells are antithrombotic and anti-inflammatory. However, when blood vessels are damaged by a viral infection, both anti-inflammatory and antithrombotic activities are lost. Then, the TF is exposed from the vascular endothelial cells, and the hemostatic system activates promptly [18]. These previous reports also suggest that fibrin formation due to hypercoagulation is involved in the aggravation of COVID-19, and that fibrin suppression is disrupted, which is consistent with the results of this study.

Considering the results of the present study, TAT might be a useful coagulation marker when assessing the coagulation/fibrinolysis system equilibrium in the early stages of COVID-19. In our hospital, TAT and PIC are routinely collected on admission; however, TAT and PIC are not routinely evaluated in general clinical settings. Therefore, since this was an exploratory study, further prospective studies are needed to support the findings of our study.

Our study has several limitations. First, this study is retrospective. Further data collection is desired in the future. Second, coagulation/fibrinolysis markers, such as TAT and PIC, were not followed up during hospitalization. COVID-19 is considered to present with suppressed-fibrinolytic-type DIC in the early and middle stages and suddenly turn into enhanced-fibrinolytic-type DIC [19]. We focused on normal D-dimer; therefore, a coagulation/fibrinolysis marker that occurs in the early stages of SARS-CoV-2 infection, it is impossible to argue at what point in time the suppressed-fibrinolytic-type DIC will transfer to the fibrinolytic-type DIC. Third, our hospital does not handle extracorporeal membrane oxygenation (ECMO) cases, and the number of deaths was low (20 of 797 patients died). This may be related to our study result that elevated PIC did not affect severe respiratory failure, even in the elevated D-dimer group. Furthermore, even in patients with elevated D-dimer where the fibrinolytic system mechanism should be functioning, the mean value of PIC was normal. This may be because Hayakawa M et al. pointed out that PIC was significantly high (20 μg/mL) in a critically severe case requiring ECMO, suggesting enhanced-fibrinolytic-type DIC [20,21]. If the facility can handle critical illnesses, such as ECMO, various patterns of PIC data may be collected.

## 5. Conclusions

Our study demonstrated that hypercoagulation due to SARS-CoV-2 infection may occur even during a normal D-dimer level, causing severe respiratory failure in COVID-19. Further prospective studies are needed to confirm these findings and validate the efficacy in general clinical settings.

## Figures and Tables

**Figure 1 jcm-11-00134-f001:**
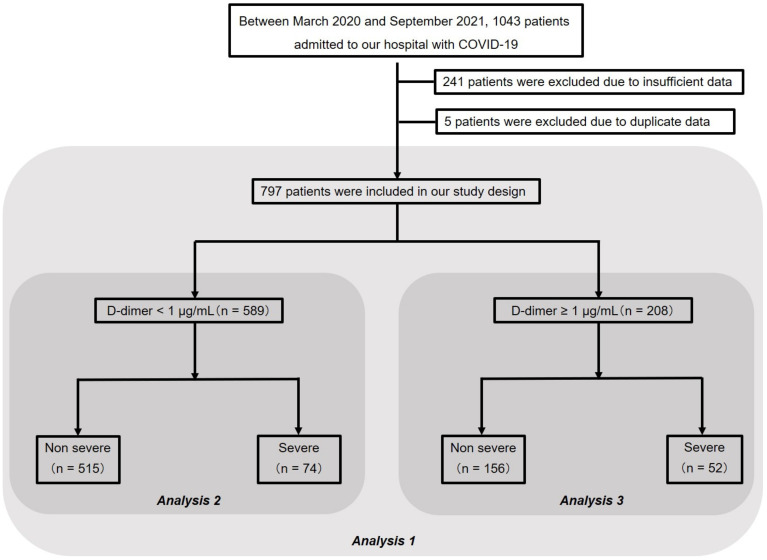
Study population flowchart. The final study cohort comprised 797 patients. Analysis 1 presents clinical characteristics of 797 patients (non-severe group, *n* = 671, and severe group, *n* = 126). More detailed results are shown in Table 1. Analysis 2 presents clinical characteristics of 589 patients (non-severe group, *n* = 515, and severe group, *n* = 74) and factors that affect severe respiratory failure in the normal D-dimer group. Analysis 3 shows clinical characteristics of 208 patients (non-severe group, *n* = 156, and severe group, *n* = 52) and factors that affect severe respiratory failure in the elevated D-dimer group. Elevated D-dimer was defined as D-dimer ≥ 1.0 µg/mL. COVID-19, coronavirus disease 2019.

**Figure 2 jcm-11-00134-f002:**
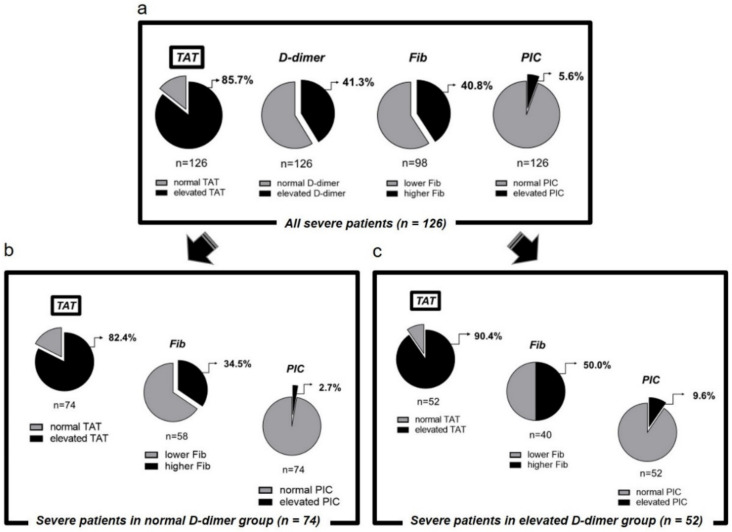
Percentage of patients in the severe group with abnormal coagulation/fibrinolytic markers. Higher fibrinogen was defined as fibrinogen ≥ 617 mg/dL, elevated D-dimer as D-dimer ≥ 1.0 µg/mL, elevated TAT as TAT ≥ 4.0 ng/mL, and elevated PIC as PIC > 0.8 µg/mL. TAT, thrombin-antithrombin complex; PIC, plasmin-alpha2-plasmin inhibitor-complex. (**a**) All severe patients. (**b**) Severe patients in normal D-dimer group. (**c**) Severe patients in elevated D-dimer group.

**Table 1 jcm-11-00134-t001:** Characteristics and outcomes of the patient cohort.

Characteristics	Non-Severe (*n* = 671)	Severe (*n* = 126)	*p* Value
Age (year)	54.8 (16.1)	55.3 (13.6)	0.7080
Male	436 (65.0%)	106 (84.1%)	<0.0001
BMI (kg/m^2^)	25.40 (4.40), NA = 26	27.35 (5.42), NA = 2	0.0002
PT (sec)	8.7 (0.9), NA = 3	9.3 (4.4), NA = 1	0.1780
APTT (sec)	30.1 (4.3), NA = 6	32.7 (6.5), NA = 1	<0.0001
Fibrinogen (mg/dL)	480.7 (137.7), NA = 163	582.2 (134.0), NA = 28	<0.0001
D-dimer (μg/mL)	1.11 (2.56)	1.39 (2.07)	0.1770
TAT (ng/mL)	6.68 (5.36)	9.88 (14.60)	0.0235
PIC (μg/mL)	0.56 (0.84)	0.70 (0.89)	0.0935

BMI, body mass index; PT, prothrombin time; APTT, activated partial thromboplastin time; TAT, thrombin-antithrombin complex; PIC, plasmin-alpha2-plasmin inhibitor-complex.

**Table 2 jcm-11-00134-t002:** Analysis of respiratory status on admission by classifying D-dimer value.

Variables	All Patients (*n* = 797)	Normal D-Dimer (*n* = 589)	Elevated D-Dimer (*n* = 208)	*p* Value
Pneumonia	703 (89.1%, NA = 8)	509 (87.3 %, NA = 6)	194 (94.2 %, NA = 2)	0.0060
SpO2 ≤ 93% on room air at sea level	21 (2.6 %)	11 (1.9 %)	10 (4.8 %)	0.0397
Conventional oxygen therapy	253 (31.7 %)	157/589 (26.7 %)	96 (46.2 %)	<0.0001
High-flow nasal cannula	41(5.1 %)	22/589 (3.7 %)	19 (9.1 %)	0.0053
Mechanical ventilation	3 (0.3 %)	0 (0 %)	3 (1.4 %)	0.0176

**Table 3 jcm-11-00134-t003:** Characteristics of patients in the normal D-dimer group.

Characteristics	Non-Severe (*n* = 515)	Severe (*n* = 74)	*p* Value
Age (year)	52.7 (15.4)	52.6 (12.0)	0.9570
Male	336 (65.2%)	65 (87.8%)	<0.0001
BMI (kg/m^2^)	25.55 (4.43), NA = 14	27.66 (5.04)	0.0002
BMI ≥ 30 (kg/m^2^, %)	77 (15.4%), NA = 14	15 (20.3%)	0.3080
Fibrinogen (mg/dL)	465.8 (132.7), NA = 130	567.7 (114.4), NA = 16	<0.0001
D-dimer (μg/mL)	0.61 (0.19)	0.73 (0.16)	<0.0001
TAT (ng/mL)	5.74 (3.70)	6.84 (3.40)	0.0165
PIC (μg/mL)	0.47 (0.46)	0.58 (0.53)	0.0565

BMI, body mass index; TAT, thrombin-antithrombin complex; PIC, plasmin-alpha2-plasmin inhibitor-complex.

**Table 4 jcm-11-00134-t004:** Factors associated with severe respiratory failure in subjects with normal D-dimer.

Variables	Univariate Analysis OR (95% CI)	*p* Value	Multivariate Analysis OR (95% CI)	*p* Value
Age (year)	-	-	1.010 (0.984–1.030)	0.6220
Male	3.841 (1.847–8.982)	<0.0001	2.440 (1.140–5.230)	0.0219
BMI ≥ 30 kg/m^2^	1.399 (0.700–2.656)	0.3080	1.330 (0.630–2.810)	0.4550
Higher fibrinogen	2.537 (1.311–4.806)	0.0038	1.860 (0.989–3.520)	0.0542
Elevated TAT	2.950 (1.554–6.011)	0.0003	2.730 (1.360–5.500)	0.0049
Elevated PIC	1.164 (0.124–5.391)	0.6920	1.670 (0.309–9.050)	0.5500

Higher fibrinogen was defined as fibrinogen ≥ 617 mg/dL, elevated TAT as TAT ≥ 4.0 ng/mL, and elevated PIC as PIC > 0.8 µg/mL. OR, odds ratio; 95% CI, 95% confidence interval; BMI, body mass index; TAT, thrombin-antithrombin complex; PIC, plasmin-alpha2-plasmin inhibitor-complex.

**Table 5 jcm-11-00134-t005:** Characteristics of patients with elevated D-dimer.

Characteristics	Non-Severe (*n* = 156)	Severe (*n* = 52)	*p* Value
Age (year)	61.7 (16.6)	59.2 (14.9)	0.3190
Male	100 (64.1%)	41 (78.9%)	0.0594
BMI (kg/m^2^)	24.87 (4.28), NA = 12	26.9 (5.95), NA = 2	0.0296
Fibrinogen (mg/dL)	527.4 (142.9), NA = 33	603.3 (157.2), NA = 12	0.0050
D-dimer (μg/mL)	2.76 (4.97)	2.33 (3.00)	0.4560
TAT (ng/mL)	9.76 (8.15)	13.83 (21.25)	0.1630
PIC (μg/mL)	0.86 (1.50)	0.86 (1.22)	0.9730

BMI, body mass index; TAT, thrombin-antithrombin complex; PIC, plasmin-alpha2-plasmin inhibitor-complex.

**Table 6 jcm-11-00134-t006:** Factors associated with severe respiratory failure in subjects with elevated D-dimer.

Variables	Univariate Analysis OR (95% CI)	*p* Value	Multivariate Analysis OR (95% CI)	*p* Value
Age (year)			0.997 (0.971–1.020)	0.8090
Male	2.080 (0.955–4.856)	0.0594	1.670 (0.665–4.180)	0.2750
BMI ≥ 30 kg/m^2^	2.699 (1.056–6.796)	0.0303	3.100 (1.180–8.150)	0.0217
Higher fibrinogen	2.946 (1.317–6.639)	0.0055	2.550 (1.150–5.650)	0.0210
Elevated TAT	1.302 (0.438–4.713)	0.8030	1.600 (0.423–6.080)	0.4880
Elevated PIC	0.616 (0.173–1.788)	0.4820	0.957 (0.277–3.310)	0.9450

Higher fibrinogen was defined as fibrinogen ≥ 617 mg/dL, elevated TAT as TAT ≥ 4.0 ng/mL, and elevated PIC as PIC > 0.8 µg/mL OR, odds ratio; 95% CI, 95% confidence interval; BMI, body mass index; TAT, thrombin-antithrombin complex; PIC, plasmin-alpha2-plasmin inhibitor-complex.

## Data Availability

The data that support the findings of this study are available from the corresponding author upon reasonable request.

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
