# Peer review of "Elevated TAT in COVID-19 Patients with Normal D-Dimer as a Predictor of Severe Respiratory Failure: A Retrospective Analysis of 797 Patients"

_jcm, 2021, doi:10.3390/jcm11010134_

Round 1

Reviewer 1 Report

Takisheta et al conducted a retrospective analysis to evaluate the association of an elevated TAT with associated D dimer values and its effects on respiratory failure which was defined as requiring the use of high-flow nasal cannula or Mechanical ventilation 

Major comments 

  • I think the introduction needs a better construct, it is extremely confusing as a reader to understand the aim as to why the study was conducted. The authors should clearly demarcate the introduction into paragraphs which include the outline of the topic background (introduce COVID very briefly), the available data thus far in a succinct fashion(paraphrasing is very important), identify the gaps in the literature followed by stating the research question, objectives and briefly introducing its components(TAT and PIC)
  • I think the methods and results sections are better written, the pie chart representation is good. However is obtaining a TAT and PIC part of routine clinical practice ? the authors must include its utility in routine practice outside of COVID 19 or this study
  • The discussion needs better wordsmithing and paraphrasing to. It clearly must delineate the findings in a paragraph, put the research findings into perspective in one to two paragraphs( why are these findings important to the bedside clinician and how could it potentially alter practice or decision making). The authors must state in the manuscript these findings are purely exploratory and need prospective studies to evaluate its utility and generalizability. 
  • I think the authors provide some good physiologic explanations and with references which is important but all this can be paraphrased better and will flow better if put into perspective as to why this study is important 

Reviewer 2 Report

In this study authors aimed to association between several coagulation/fibrinolysis markers measured at admission and respiratory failure. They studies thrombin-antithrombin complex (TAT), plasmin-alpha2-plasmin inhibitor-complex (PIC), D-dimer and fibronogen levels. At the end they found that an elevated TAT within normal D-dimer and higher fibrinogen with elevated D-dimer were risk factors for severe respiratory failure.

The manuscript is presented in a well-structured manner. The topic is novel, interesting and relevant. The design and methodology are clear. Tables are figures are adequate. Ethics and data availability statements are adequately presented. And finally discussion is well written.
